# DNA Damage Response Alterations in Ovarian Cancer: From Molecular Mechanisms to Therapeutic Opportunities

**DOI:** 10.3390/cancers15020448

**Published:** 2023-01-10

**Authors:** María Ovejero-Sánchez, Rogelio González-Sarmiento, Ana Belén Herrero

**Affiliations:** 1Institute of Biomedical Research of Salamanca (IBSAL), 37007 Salamanca, Spain; 2Molecular Medicine Unit, Department of Medicine, University of Salamanca, 37007 Salamanca, Spain; 3Institute of Molecular and Cellular Biology of Cancer (IBMCC), University of Salamanca-Spanish National Research Council, 37007 Salamanca, Spain

**Keywords:** ovarian cancer, DNA damage response, DNA repair, direct reversal repair, mismatch repair, nucleotide excision repair, base excision repair, homologous recombination, nonhomologous end joining, ATM, ATR, p53

## Abstract

**Simple Summary:**

The DNA damage response (DDR) is frequently altered in ovarian cancer (OC), which can be exploited for therapeutic purposes. Moreover, targeting DDR signaling pathways has become an attractive strategy for increasing the effect of DNA-damaging drugs and overcoming chemoresistance. Here, we review the main DDR pathways and their alterations in OC. We also recapitulate the preclinical and clinical studies that target the DDR for the treatment of the disease.

**Abstract:**

The DNA damage response (DDR), a set of signaling pathways for DNA damage detection and repair, maintains genomic stability when cells are exposed to endogenous or exogenous DNA-damaging agents. Alterations in these pathways are strongly associated with cancer development, including ovarian cancer (OC), the most lethal gynecologic malignancy. In OC, failures in the DDR have been related not only to the onset but also to progression and chemoresistance. It is known that approximately half of the most frequent subtype, high-grade serous carcinoma (HGSC), exhibit defects in DNA double-strand break (DSB) repair by homologous recombination (HR), and current evidence indicates that probably all HGSCs harbor a defect in at least one DDR pathway. These defects are not restricted to HGSCs; mutations in *ARID1A*, which are present in 30% of endometrioid OCs and 50% of clear cell (CC) carcinomas, have also been found to confer deficiencies in DNA repair. Moreover, DDR alterations have been described in a variable percentage of the different OC subtypes. Here, we overview the main DNA repair pathways involved in the maintenance of genome stability and their deregulation in OC. We also recapitulate the preclinical and clinical data supporting the potential of targeting the DDR to fight the disease.

## 1. Introduction

Ovarian cancer (OC) includes diverse tumors that affect the ovaries, the fallopian tubes, or the peritoneal cavity. Most (90%) present with an epithelial origin and have been classically divided into five histological subtypes: high-grade serous carcinoma (HGSC), low-grade serous carcinoma (LGSC), endometrioid carcinoma, clear cell (CC) carcinoma, and mucinous carcinoma [1,2,3,4,5]. Each subtype exhibits distinct genetic alterations and clinical and prognostic characteristics, which are summarized in Table 1.

OC represents the nineth most common type of cancer, the eighth leading cause of cancer death in women, and the most lethal gynecologic malignancy [6,7,8]. This high mortality is mainly due to asymptomatic tumor growth, which results in a late diagnosis [9]. Moreover, disease relapse is quite common after surgery and standard platinum/taxane-based chemotherapy [10,11], and after further treatments with different chemotherapeutic agents [12]. Therefore, there is a clear need to develop new therapeutic strategies that should ideally target patient-specific genome alterations, the so-called precision medicine. One of these strategies could be novel therapies targeting the DNA damage response (DDR), which is affected in many ovarian tumors.

## 2. The DNA Damage Response (DDR)

Mammalian genomes are constantly being assaulted by endogenous and exogenous DNA-damaging agents. To fight this threat, cells have evolved the DDR, a collection of interdependent signaling pathways that detect the lesions, signal its presence, and mediate its repair [13]. Defects in these mechanisms cause human diseases, such as cancer; however, defective cells generally display higher sensitivity towards DNA-damaging drugs, which may convert such deficiencies into their Achilles’ heel [14,15]. 

The most prominent sources of endogenous damage are reactive oxygen species (ROS), which are mostly formed as a byproduct of mitochondrial respiration [16,17]. Another physiological process, DNA replication, may introduce some DNA mismatches that need to be repaired [18]. Additional endogenous sources of DNA lesions are byproducts of lipid peroxidation, endogenous alkylating agents, and reactive carbonyl species, as well as spontaneous hydrolysis of DNA, which results in apurinic/apyrimidinic abasic sites [19,20,21]. On the other hand, the most important sources of exogenous DNA damage include ultraviolet (UV) light, ionizing radiation (IR), chemotherapeutic agents, and other environmental carcinogens that could be inhaled or ingested [22,23] (Figure 1).

These different DNA-damaging agents create a diversity of DNA lesions that activate the DDR. The full system comprises three sets of proteins, namely the sensors of DNA damage, the transducers, and the effectors (Figure 1). The sensor proteins actively search the genome for the presence of DNA damage. If damage exists, they transmit a signal to other proteins called transducers, such as ATM and ATR kinases. These transducers recruit and activate effector proteins, such as CHK1, CHK2, and p53, that activate cell cycle checkpoints allowing DNA repair or triggering apoptosis or senescence. The goal is to avoid the transmission of erroneous genetic information to daughter cells [13]. 

The type of DNA lesion determines the repair mechanism employed by the cells (Figure 2). DNA alkylation produced by endogenous metabolites or alkylating drugs is repaired by the direct reversal mechanism (DR) [24]; replication errors are corrected by the mismatch repair (MMR) pathway [25]; bulky adducts caused by a variety of DNA-damaging agents, including pyrimidine dimers produced by UV exposure, are repaired by nucleotide excision repair (NER) [26]; single-strand breaks (SSBs) and small base damage caused by endogenous or exogenous sources can be repaired by base excision repair (BER) [26]; Finally, double-strand breaks (DSBs) produced by IR, ROS, or some chemotherapeutics are solved by homologous recombination (HR) or nonhomologous end joining (NHEJ) repair pathways [27].

These core DNA repair pathways may be interconnected and, under certain circumstances, failure to repair by one of the pathways can be compensated by the activity of others. Moreover, all these repair pathways do not function in isolation, but they are integrated with complementary processes that are essential to overall genome maintenance. These processes, which also form part of the DDR, as mentioned before, include cell cycle checkpoints, the p53 pathway, and chromatin remodeling factors that facilitate the accessibility of repair proteins.

The following sections describe the main DDR pathways, the proteins involved, and their alterations in OCs, which is summarized in Table 2 and Table 3. The potential of targeting these pathways in OC treatment is also overviewed and summarized in Table 4 (preclinical studies) and Table 5 (clinical studies). The information shown in the tables was found using Google Scholar (https://scholar.google.es accessed on 31 October 2022) or PubMed (https://pubmed.ncbi.nlm.nih.gov accessed on 31 October 2022) databases. Search terms included: “DNA damage response”, “DNA repair”, or the names of the different DDR pathways and compounds together with the term “ovarian cancer”. ClinicalTrials database (https://beta.clinicaltrials.gov accessed on 30 November 2022) was also employed to obtain information concerning clinical studies. The name of each compound and “ovarian cancer” were used as search terms. Scientific literature derived from each clinical trial was also searched using Google Scholar or PubMed.

## 3. DNA Repair Pathways and Their Alteration in OC: Implications for Therapy

Alterations of DNA repair pathways, caused by genetic inactivation and/or epigenetic mechanisms, represent a common feature of carcinogenesis; they drive malignant transformation by the accumulation of genomic alterations in the cells [219]. Defects in DNA repair can occur at the germline, conferring an increased risk of developing cancer, or can be somatic, which might also result in sporadic cancers or affect sensitivity to therapy. 

### 3.1. Direct Reversal Repair (DR)

During normal metabolism, or as a result of exposure to various carcinogens or alkylating agents, the nitrogenous base guanine can become methylated. The enzyme methylguanine DNA methyltransferase (MGMT) deals with this damage by transferring the methyl group at the 6-position of guanine to a cysteine group located in the MGMT active center. Then, the enzyme is irrevocably inactivated, which is the reason why MGMT is known by the term ‘suicide protein’ [28]. This repair system is called direct repair because it acts without the need to excise the DNA helix (Figure 2). 

Loss of MGMT expression has been described in many tumor types, including glioblastoma, lymphoma, breast and prostate cancer, and retinoblastomas, and its usually due to promoter methylation [220]. Interestingly, defects in direct repair by MGMT have been linked to the therapeutic success of alkylating agents, especially temozolomide (TMZ). Thus, in glioblastomas, where approximately 45% bear a methylated MGMT promoter, treatment with TMZ produces a survival advantage for these types of patients [221] and is used as an oral alkylating agent to treat the disease.

In ovarian cancer, Roh et al. [51] detected MGMT promoter hypermethylation in 14% of the samples analyzed (86 epithelial OCs). In addition, a meta-analysis including 10 studies and 910 OC samples concluded that MGMT-inactivation might be associated with carcinogenesis in certain histological types (non-serous carcinomas) [52] (Table 3). Additionally, a role for MGMT in the chemoresistance of ovarian cancer has been recently described by Wu et al. [222]. The authors found that the deubiquitinating enzyme 3 (DUB3) stabilized the anti-apoptotic protein MCL1 and that MGMT was a key activator of DUB3 transcription. Consequently, the MGMT inhibitor PaTrin-2 effectively suppressed OC cells with elevated MGMT-DUB3-MCL1 expression. This inhibitor, known as Lomeguatrib, has also proven to sensitize cells from advanced solid tumors to temozolomide [97] (Table 4) and was tested in clinical trials to determine the optimal doses in patients with several tumors, including OC [171] (Table 5).

### 3.2. Mismatch Repair (MMR)

The DNA mismatch repair system corrects spontaneous base–base mispairs and small insertions–deletion loops (indels) generated by failures in DNA replication. It is therefore one of the most important guardians of genome integrity. Defects in MMR increase the mutational rate of the cell and alter the sequence lengths within microsatellites, which is called microsatellite instability (MSI) [18]. The first step in this repair pathway is the recognition of the lesion. The most abundant mismatch-binding factor is composed of the ATPases MSH2 and MSH6, which recognizes single base substitutions or small insertion or deletion loops (IDLs) [29,30,31] (Figure 2). The repair of larger IDLs is initiated by the MSH2–MSH3 complex. Upon damage recognition, these complexes bind to the mispairing site and recruit another complex formed by MLH1 and PMS1 or PMS2. This complex has endonuclease activity and, in the presence of ATP, excises the DNA chain at the error site. The stabilizing protein RPA allows the binding of proliferating cell nuclear antigen (PCNA) and replication factor C (RFC) to protect the gap generated in the DNA. Subsequently, DNA exonuclease I (Exo1) enters into the DNA structure guided by the two complexes (MSH2–MSH3/MSH6 and MLH1–PMS1/PMS2) and removes the damaged area along with other nearby nucleotides. At this point, the DNA polymerase (Pol δ) synthesizes DNA in the deleted region, thus, errors that escaped polymerase proofreading in the first place will finally be resynthesized again by Pol δ as part of mismatch repair [32]. Finally, PCNA factor checks the synthesis of nucleotides, and DNA ligase I seals the nick [33].

Defects in the MMR are associated with Lynch syndrome, an autosomal dominant syndrome that mainly predisposes to colon cancer, but also to endometrial and ovarian cancer [60,223]. In Lynch syndrome, most mutations occur in *MLH1* (42%) and *MSH2* (33%), followed by *MSH6* (18%) and *PMS2* (7.5%) [53]. Patients with this hereditary condition usually acquired only one mutated allele from one of the progenitors and lose the second allele somatically via mutation or methylation. Lynch syndrome confers a 10–15% risk of developing low-grade or clear cell OC, which tends to develop at an early age [54,55]. HGSC seems little influenced by defects in MMR, as revealed by a large study of 2222 ovarian cancer cases that found defective MMR in only 17 cases [224]. 

In sporadic cancers, alterations in the MMR pathway have also been described. The most common finding is promoter hypermethylation of the *MLH1* gene, which leads to its silencing, and has been observed in sporadic MSI-cancers including colorectal, endometrial, and ovarian cancer [56]. The frequencies of *MLH1* promoter hypermethylation ranged between 10% and 50%, with the higher estimates reported in MSI-tumors [57,58].

It is known that MMR-deficient cells are resistant or acquire resistance to common chemotherapeutic drugs, such as 5-FU, used against colorectal cancer, or cisplatin and carboplatin, widely used for the treatment of ovarian cancers [225,226]. This fact, together with the significant frequency of MMR-deficient tumors, highlighted the need to identify new therapeutic strategies that target vulnerabilities exhibited by these malignant cells. Synthetic lethal approaches have been explored with the aim of killing MMR-deficient cells and several candidates have been identified in preclinical studies [225]. However, none of them have yet been explored in clinical studies with OC patients. Therefore, although various laboratory methods exist to identify MMR-deficient ovarian cancers, such as microsatellite instability analysis, immunohistochemistry, promoter hypermethylation testing, and germline mutation analysis, strategies to specifically target those tumors are not yet available. It has been reported, however, that defects in MMR create a ‘mutator phenotype’ that results in the synthesis of “non-self” immunogenic antigens, which increase the susceptibility to immune checkpoint inhibition [227]. That seems to be the explanation for the good response of different MMR-deficient tumors to single-agent PD-L1. Indeed, the US Food and Drug Administration (FDA) has approved pembrolizumab, one immune checkpoint inhibitor, for any MMR-deficient relapsed solid tumor. This indication is not yet available in Europe but might orient trials with relapsed endometrioid or clear cell OCs and MMR deficiency [228].

### 3.3. Nucleotide Excision Repair (NER)

NER eliminates various bulky (helix-distorting) lesions, such as those produced by ultraviolet light (UV), certain environmental chemical mutagens, or the inter- and intra-strand crosslinks induced by chemotherapeutic agents like cisplatin [34,229] (Figure 2).

Two NER sub-pathways have been described: transcription-coupled NER (TC-NER) and global genome NER (GG-NER). They differ in the damage-detection mechanisms but utilize the same machinery to excise and repair the damage [34]. TC-NER repairs the lesions present in the transcribed strand of active genes. It is initiated by the RNA polymerase stalled at the lesion together with TC-NER-specific factors CSA, CSB, and XAB2. GG-NER repairs the lesions that occur anywhere in the genome and is initiated by the GG-NER-specific factor XPC-RAD23B, in some cases with the help of UV-DDB (UV-damaged DNA-binding protein) [26].

Once the damage has been detected, the XPA/RPA and TFIIH complexes are recruited and remain bound to the DNA. The TFIIH complex, formed by different helicases such as XPB and XPD, is responsible for opening and stabilizing the DNA helix. Next, two endonucleases (XPG and XPF/ERCC1) cleave on both sides of the lesion, eliminating several nucleotides. After excision, the resulting gap of approximately 30 nucleotides is filled in by DNA synthesis and ligation. This process is carried out by the action of replication factors (PCNA and RFC); DNA polymerases δ, ε, and κ; and DNA ligases I and III [34,35].

Defective NER is characteristic of the skin cancer-prone inherited disorder xeroderma pigmentosum, which is characterized by extreme UV sensitivity, but might also occur in other sporadic cancers [230]. Thus, gene polymorphisms in some NER proteins have been associated with lung, skin, and bladder cancers [60,230].

In ovarian cancer, one study detected an association between some SNPs in NER proteins and ovarian cancer susceptibility [59]. However, the most significative connection between NER and OC came from the TCGA data set, which revealed that 8% of HGSC OCs harbored alterations in some NER genes, which included homozygous deletions, missense, or splice site mutations [60,61]. 

As mentioned previously, the NER signaling pathway is involved in the repair of platinum-induced adducts; therefore, NER deficiency may increase sensitivity to cisplatin, whereas NER upregulation might mediate cisplatin chemoresistance. In this regard, Ceccaldi et al. showed that patients with HGSC tumors bearing NER gene mutations displayed improved survival to platinum compared to those that did not bear NER alterations [61]. Moreover, two NER mutations (ERCC6-Q524 and ERCC4-A583T), which have been identified in the two most sensitive tumors, were functionally associated with platinum sensitivity *in vitro*. On the other hand, it has been recently described that the tyrosine kinase receptor TIE-1 mediates OC platinum resistance by promoting NER [231]. 

Based on all these data, the development of NER inhibitors could represent a therapeutic strategy against NER-deficient or platinum-resistant OCs. Currently, there are no targeted therapies approved by the FDA specifically for patients with germline or somatic mutations in NER pathways genes. However, several small inhibitors have been developed during the last years with a variable potency [60,232,233]. Certain NER inhibitors, such as MCI13E, TDRL-505, or TDRL-551, have shown antitumor activity in OC cells [98], and TDRL-551, an inhibitor of RPA protein, produces a synergistic cytotoxic effect when combined with platinum or etoposide in OC cells [98] (Table 4).

Another promising group of compounds with anti-tumor activity is the ecteinascidin family, which includes trabectedin and lurbinectedin [99]. These compounds interfere with the NER machinery, attenuating the repair of certain NER substrates, and use NER proteins to exert their cytotoxic effect [100,234]. Preclinical studies have shown that lurbinectedin and trabectedin were effective in the treatment of cisplatin-sensitive and cisplatin-resistant OC cells and xenograft tumor models, especially when they were combined with cisplatin [99,100,101,102]. In addition, Casado et al. have suggested that trabectedin could re-sensitize tumor cells to platinum therapy [235]. The combination of lurbinectedin/trabectedin with other antineoplastic drugs, such as doxorubicin or irinotecan, also exerted a synergistic effect in OC cells [236,237]. Its combination with pegylated liposomal doxorubicin (PLD) improved progression-free survival and overall survival over PLD alone in patients with recurrent OC [238,239,240]. Indeed, trabectedin together with PLD is indicated for the treatment of patients with relapsed platinum-sensitive OC [241]. Based on these preclinical studies, several clinical trials have recently explored the effectivity of trabectedin and lurbinectedin in monotherapy and in combination with PLD or other drugs [172,173,174,175,176,177,178,179,180,181,182,189,190,191,192,193] (Table 5). 

### 3.4. Base Excision Repair (BER)

BER corrects small base lesions (alkylations, oxidations, deaminations, depurinations) or single-strand breaks (SSBs) resulting from endogenous or exogenous sources, such as radiation or chemotherapeutic agents [36]. This repair pathway is initiated by a DNA glycosylase that recognizes and removes the damaged base, leaving an abasic site that is further processed in several steps: incision (which creates a SSB), end processing, DNA synthesis, and ligation. There are 11 families of glycosylases responsible for detecting the different types of lesions, for example, OGG1, UNG, or MUTYH. Several polymorphisms in these glycosylases have been linked with various cancers, such as colorectal, oesophageal, gastric, or lung cancer [62,242]. In the case of colorectal cancer, it has been described a predisposition syndrome that is associated with biallelic-inherited mutations of *MUTYH* [243]. 

The main endonuclease responsible for the incision step is APE1 that generates a SSB in the DNA. SSBs formed by the action of this enzyme, or those directly created by endogenous or exogenous sources, are rapidly detected and bound by PARP1. This protein stabilizes the DNA ends and adds a poly-ADP-ribose chain that recruits downstream proteins such as XRCC1. XRCC1 serves as a scaffold that attracts other proteins required for the repair, specifically, DNA ligase III (LIG3), DNA polymerase β, and bifunctional polynucleotide kinase 3′-phosphatase (PNKP) [37,38] (Figure 2).

The implication of BER defects in the development of OC is not clear. However, some studies that have reported a connection, such as those that associate several polymorphisms in the glycosylase OGG1, the endonuclease APE1, or the XRCC1 protein with an increased risk of OC [62,63,64,65,66,67,68]. High nuclear expression of the endonuclease APE1 has also been associated with the development of HGSC, and a correlation with worse overall survival and greater resistance to platinum therapy was reported [69]. In another study, APE1 overexpression was described to promote OC progression. Indeed, the authors found that APE1 downregulation inhibited ovarian cancer cell proliferation, which pointed out this protein as a potential therapeutic target [244]. Moreover, APE1-knockdown cells showed a stronger apoptosis induction after being exposed to DNA-damaging agents, such as UV, camptothecin, or cisplatin [244,245]. Therefore, the use of DNA damage agents together with an APE1 inhibitor could represent a therapeutic strategy. Several APE1 inhibitors have been developed, such as methoxyamine or E3330 (Table 4). Methoxyamine (TRC102) has been described to enhance temozolomide cytotoxicity in several OC cell lines by increasing the amount of DNA damage and apoptosis [103]. This combined treatment was tested in clinical trials for several solid tumors, including granulosa cell OC [195] (Table 5). The E3330 inhibitor and several analogs have shown to inhibit OC cell proliferation [104,105]. Spiclomazine and fiduxosin, two inhibitors of APE1/NPM1 interaction, have also been described to decrease proliferation and sensitize OC cancer cells to bleomycin [106]. 

The most important BER inhibitors used in the clinic are those that target PARP1. PARP inhibitors (PARPi) were first approved for the treatment of breast and ovarian cancers with defects in the HR pathway due to *BRCA1* or *BRCA2* mutations. The synthetic lethality between PARP inhibition and BRCA inactivation was first reported in 2005, where it was proposed that inhibition of PARP1 activity would lead to replication fork collapse and the subsequent formation of DSBs, lesions that require the HR pathway to be repaired [107,108]. New studies have shown that, in addition to women with *BRCA*-mutated tumors, initial treatment with PARP inhibitors also benefits other OCs with defects in HR, which is a major step forward. 

There are three PARP inhibitors approved for use in OC [246,247]: olaparib, rucaparib, and niraparib. Olaparib was the first PARPi approved for clinical use. It currently has two indications in patients with advanced OC. The first is for treatment in patients with a mutation or suspected germline mutation in *BRCA1/2* after three or more prior lines of chemotherapy. The second is as maintenance treatment in patients with recurrent OC who are in partial or complete response to platinum-based chemotherapy [246,247]. Rucaparib was the second PARPi approved for OC treatment and presents two indications in OC: maintenance treatment in patients with recurrent OC who are in complete or partial response to platinum-based chemotherapy, and the treatment of patients with pathogenic *BRCA* mutations (germline or somatic) associated with this tumor, who have undergone two or more lines of chemotherapy [246,247]. Finally, niraparib is indicated for the maintenance treatment of patients with recurrent OC who have a partial or complete response to platinum-based chemotherapy [246,247]. 

The clinical trials exploring the PARPi olaparib, rubaparib, niraparib, talazoparib, and pamiparib in monotherapy or in combination with other drugs are shown in Table 4 and Appendix A. Many of them are DDR-interfering drugs, such as DNA-damaging agents (temozolomide, carboplatin, cisplatin, or pegylated liposomal doxorubicin) or inhibitors of DNA damage response (ATR inhibitors, WEE1 inhibitors, or BET inhibitors). These combinations are supported by recent evidence that suggests that PARP1 may also have a role in other DNA repair pathways, including NER, MMR, and DSB repair, the pathway described in the next section [248,249,250]. 

Several studies have described that those tumors bearing mutations in BER genes had increased neo-antigen production and PD-L1 expression [251]. Therefore, the combination of PARPi with immune checkpoints inhibitors such as pembrolizumab, dostarlimab, durvalumab, tremelimumab, atezolizumab, or avelumab, has also been studied. Moreover, PARP inhibitors have been combined with angiogenesis inhibitors such as bevacizumab, everolimus, or surufatinib. These combinations have been tested because angiogenesis and PARP inhibitors are indicated as front-line or as maintenance treatment for OC patients. 

### 3.5. DNA Double-Strand Break Repair by Homologous Recombination (HR)

DSBs are formed following exposure to ionizing radiation or some chemotherapies, and by the action of free radicals produced by endogenous metabolism [252]. This type of lesion represents a major threat to genome stability since it can lead to profound genome rearrangements [253]. Consequently, inherited, or somatic defects in DSB repair increase cancer susceptibility. Specifically, inherited defects predispose patients to cancer, especially OC and breast cancer. DSBs are repaired by two main pathways in mammalian cells: the efficient, but error-prone, nonhomologous end-joining (NHEJ) or the less efficient, but error-free, homologous recombination (HR). In case of defective NHEJ or HR, alternative (Alt)-NHEJ provides a backup mechanism where PARP1 is also involved [248].

The HR pathway is high-fidelity because it uses the homologous sister chromatid as a template to repair the lesion, and consequently, it is restricted to S and G2 phases of the cell cycle, when DNA has been replicated. The first step of the pathway is to process the ends of the DSB by nucleolytic resection. This is carried out by the MRN complex, consisting of MRE11, RAD50, and NSB1 [39,40]. The complex binds to both sides of the DSB and activates the kinase ATM, which controls cell cycle checkpoints, arresting the cell cycle, and recruiting a larger number of DNA repair proteins, including BRCA1. This protein promotes end resection and participates in HR repair at multiple stages [41]. The DNA end-resection mechanisms lead to the formation of 3′-tailed ssDNA. ATR kinase can be activated by these ssDNA intermediates controlling, therefore, the later steps of HR. The ssDNA ends are then coated by RPA, a protein that protects them from the action of nucleases. BRCA1 also forms complexes with other proteins such as the PALB2, which in turn binds to BRCA2 and enables RAD51 filament formation that replaces RPA [42]. RAD51 is a recombinase; it facilitates the invasion of the sister chromatid allowing the formation of a D-loop structure that leads to repair. Finally, the DNA polymerase and the DNA ligase definitively repair the damage [39,40]. Together with the mentioned proteins, many others have also been involved in the global HR process [40]. 

It is estimated that approximately 50% of OCs harbor some HR deficiency [73]. Germline mutations in the *BRCA1/2* genes are the most frequent and well-known mechanisms and appear in approximately 20% of HGSCs (Table 1). However, germline or somatic mutations in other HR genes and epigenetic modifications have also been implicated in OC [70,71,72]. Starting with the MRN complex, associations between mutations in MRN complex genes and OC susceptibility have been observed [74]. Moreover, an immunohistochemical study revealed that 41% of epithelial low-grade OC lacked the MRN complex and 10.3% of tumors lacked RAD50, specifically [75]. Mutations in RAD51C or RAD51D have also been associated with an increased risk of OC, having potential use in routine clinical genetic testing [76]. Regarding epigenetic modifications, the promoter region of the *RAD51* gene has been found hypermethylated in 3% of HGSC patients, leading to a deficiency in HR [60]. 

Considering that HR deficiency is a major hallmark of OC, considerable efforts are being dedicated to specifically target those defects. As mentioned previously, a synthetic lethal strategy by targeting PARPi in HR-deficient OCs has attracted great attention, in view of its favorable clinical result; however, treatment with PARPi could benefit not only *BRCA1/2* carriers, but also other OCs with HR deficiencies. For example, Zhang et al. reported that 18% of *BRCA* WT OC patients (from a total of 220) exhibited *RAD50* deletions, which were also associated with better OS and PFS with PARP inhibitors [77]. In fact, Mukhopadhyay et al. found that all HR defective OCs, identified by a RAD51 immunofluorescence assay, were more sensitive to PARPi in vitro and showed enhanced clinical platinum sensitivity [254].

There are some drugs able to inhibit HR that could also be combined with many other molecules that induce DSBs to increase the cytotoxic effect. Consequently, several works have combined genotoxic agents with DNA repair inhibitors in vivo and have found cytotoxic effects in tumor cells and lower toxicity in normal cells [255,256,257]. In this regard, it has been observed in many clinical cases that *BRCA* WT patients also respond to DNA damage/repair targeted therapeutic drugs. These results might be explained by the presence of other defects in HR independent of *BRCA1/2* mutations, as mentioned before, by other DDR defects, or by the increased proliferation rate that sensitizes tumor cells to DNA damage. In this regard, our group has recently described that the antimalarial drug chloroquine induces DNA DSBs in OC, and its combination with Panobinostat, a histone deacetylase inhibitor (HDACi) that inhibits HR [109], or NHEJ inhibitors, synergistically induce OC cell death [109,116]. Whether these combinations are more effective in HR-deficient cells needs to be further explored. Panobinostat treatment has been tested in clinical trials showing potential anticancer activity against OC [196].

Another HR inhibitor investigated is mirin, which inhibits Mre11 [258]. It has been described that this compound increased sensitivity to DNA-damaging agents such as cisplatin, carboplatin, or chloroquine in OC cells [109,110,111]. Other compounds present in the pomegranate extract have also been reported to downregulate the MRN complex and some genes involved in HR repair [258]. The authors found that the main components of pomegranate (ellagic acid and luteolin) reduced the proliferation and migration of OC cells [115]. In addition, this compound inhibited tumor growth in xenograft models of OC [115] (Table 4). 

### 3.6. DNA Double-Strand Break Repair by Nonhomologous End Joining (NHEJ)

NHEJ is the other pathway that deals with DNA DSBs. This pathway is considered fast and efficient; however, it may cause the loss of some nucleotides on both sides of the break or an alteration of the base pair sequence at the break site [43,44]. In general, this loss or alteration is not critical for the cell, since the genome is rich in repetitive sequences; however, both downregulation and upregulation of the pathway can lead to genome instability [259]. NHEJ repair can take place in all phases of the cell cycle, although it has low activity in the S and G2 phases [43,44]. Two NHEJ sub-pathways have been described, the canonical and the alternative pathway [43].

The canonical NHEJ repair pathway requires the activity of different proteins: Ku70, Ku80, DNA-dependent kinase catalytic subunit (DNA-PKcs), Artemis, DNA ligase IV, XRCC4, and XLF (Figure 2). When the break is detected, the heterodimer Ku70/Ku80 binds to DSBs with high affinity, protecting the DNA from the action of the exonucleases and serving as a scaffold to recruit the other NHEJ factors to the damage site, such as DNA-PKcs. This protein is a member of phosphatidylinositol-3 (PI-3) kinase-like kinase family (PIKK), which also includes two other kinases important in the cellular response to DNA damage, ATM and ATR. If the DNA ends of the break are not compatible, they must be trimmed, a process that is carried out by the nuclease Artemis. Once the ends are compatible, ligation is carried out by a complex consisting of XRCC4, XLF, and DNA ligase IV [43,44] (Figure 2). 

The alternative NHEJ repair pathway is less characterized but is considered an important promoter of cancer genome instability [45]. It is independent of the heterodimer Ku70/Ku80 and employs regions of microhomologies, which may be distant from the breakpoint, to repair the lesion. The pathway is therefore associated with the loss of genetic material and is highly mutagenic. Several studies propose the role of Alt-NHEJ as a backup pathway when C-NHEJ or HR are defective [45,46]. The key proteins in this pathway are PARP1, DNA ligase III, and XRCC1, and it is believed that the process is promoted by DNA polymerase Θ. During the repair, PARP1 recognizes the double-strand break and binds to DNA in competition with Ku70/80. XRCC1 and DNA ligase III form a complex that is responsible for binding the double-strand breaks. 

In OC, one study carried out by McCormick et al. described that the NHEJ repair pathway was altered in up to 50% of tumors independent of the HR repair. Dysregulation of this pathway can be due to mutations in the genes involved, both in germline and somatic [60], or to their overexpression. For example, elevated expression of DNA-PKs is a common finding in HGSC and correlates with an advanced stage of the disease, as well as with a high grade, worse survival, and reduced platinum sensitivity [78]. Another example is the overexpression of DNA polymerase Θ, which promotes Alt-NHEJ pathway, has been identified in ovarian serous carcinomas [79]. Overexpression of XRCC4 has been linked to poor outcome in OC and has been proposed as a candidate biomarker for OC [80]. On the other hand, single nucleotide polymorphisms (SNPs) in the DNA ligase IV or *XRCC1* gene have been found in some OCs. These variants might affect DNA repair or sensitivity to platinum treatments [60]. 

In recent decades, diverse DNA-PKs inhibitors have been developed. They increase radio- and chemosensitivity and differ in their potency and selectivity [260]. The inhibitors NU-7026 and NU-7441 share a common chemical structure; however, NU-7441 showed a higher potency in inhibiting DNA-PKs. Both compounds have proven their efficacy in enhancing the cytotoxicity of agents that damage DNA, such as irradiation, chloroquine, or cisplatin, in ovarian cancer cell lines [116,117,118]. Peposertib (M3814), an oral DNA-PKs inhibitor, increased radiosensitivity and sensitized tumor cells to different chemotherapeutic drugs [119], such as etoposide [120,256]. Moreover, it also increased cytotoxicity to pegylated liposomal doxorubicin in a xenograft OC model [256]. This combination is being studied in human clinical trials for patients with recurrent HGSC and LGSC ovarian carcinomas [261] (Table 5). Another DNA-PKs inhibitor is AZD7648, which increased cytotoxicity to irradiation and to several chemotherapeutic drugs such as doxorubicin or PARP inhibitors in tumor cell lines including OC cells [121,255]. Moreover, this inhibitor has been proven to sensitize OC patient-derived xenografts to PLD and olaparib and to prevent abdominal metastases [122]. According to these data, the combination of DNA-PK inhibitors and a DNA-damaging agent should be considered for further preclinical and clinical studies due to their therapeutic potential. 

## 4. DDR-Associated Pathways and Their Alteration in OC: Implications for Therapy

As mentioned before, in addition to core DNA repair pathways, other DDR-associated elements exert an essential role in maintaining DNA integrity. These include chromatin remodeling factors, which enable access to DNA damage, the checkpoint signaling, and p53 pathway, which allow time for repair preventing cells from entering mitosis with substantial unrepaired damage.

### 4.1. Chromatin Remodelers

DNA in eukaryotes is highly compacted with the help of histone proteins forming the chromatin. Therefore, DNA damage repair requires opening of the chromatin structure to facilitate the accessibility of DNA repair proteins. Modification of chromatin occurs via two mechanisms: posttranslational modification of histones or displacement of these proteins, this last requires the action of ATP-dependent chromatin remodeling complexes and histone chaperones. Ubiquitination is an example of histone posttranslational modification important in the DDR. It has been described that this modification changes the chromatin structure in the vicinity of DSBs and serves as a platform to select and recruit repair proteins [262] 

The ATP-dependent mechanisms use the energy of ATP hydrolysis to disrupt the DNA–histone contacts and constitute most of the remodeling activity in the cell [47]. Four families of multi-subunit chromatin remodeling complexes have been described: SWI/SNF, INO80, CHD, and ISWI. They differ in their epigenetic reader domains, which recognize the specifically modified histone tails.

It is becoming increasingly clear that ATP-dependent chromatin remodeling complexes play important roles in the establishment and progression of human cancers, which is due, at least in part, to their role in the DDR. Somatic mutations and deregulated expression of several subunits of chromatin remodeling complexes have been described in many cancer subtypes [263], being subunits of the SWI/SNF family the most frequently altered. For example, mutations in *INI1* and *ARID1A* components, both involved in the DDR [264], are frequently observed in several tumors from diverse tissues including the stomach, large intestine, central nervous system, bone, endometrium, liver, urinary tract, and ovary [263]. Interestingly, around 30% of endometrioid OCs display mutations in *ARID1A* [81,82], which has also been found mutated in 50% of CC carcinomas [81,82,83,84]. ARID1A is considered a tumor suppressor through multiple mechanisms that include transcriptional regulation, cell cycle control, replication stress, and DNA repair. ARID1A has been described to play a role in DSB repair by helping the recruitment of the ATPase subunit of the SWI/SNF complex to DNA damage sites [265]. In addition, it also helps to recruit the NHEJ factors Ku70/Ku80 to the DSB sites and maintains checkpoint signaling through its interaction with ATR [265]. Moreover, a putative role of ARID1A in MMR has also been described. In summary, ARID1A protects the genome by interacting with the proteins of different DNA repair mechanisms. Its inactivation in an important number of OCs makes it a good candidate for synthetic lethal targeting [265]. 

The chromatin remodeling complex INO80 is also recruited to DSBs and is needed for efficient repair by HR and probably by NER [266]. Mutations in INO80 subunits are not abundant in human cancers; however, several INO80 subunits have been found overexpressed. In the case of OC, amplification of the ACT6LA subunit have been detected, which correlates with platinum chemoresistance [88,90]. 

Regarding CHD chromatin remodeling complexes, mutations in *CHD5* and overexpression of CHD8 subunits have been reported in OCs [88]. CHD4, which forms the nucleosome remodeling and deacetylase (NuRD) complex, is overexpressed or mutated in some OCs [85]. Thus, Zhao et al. [86] identified 11 out of 52 patients that exhibited a heterozygous somatic *CHD4* mutation, and Le Gallo et al. [87] also reported a somatic mutation in *CHD4* in 17% of patients with serous endometrial cancer. In addition, it has been described that *BRCA2*-mutant ovarian cancers with reduced CHD4 expression significantly correlate with shorter progression-free survival and shorter overall survival [267]. On the other hand, *CHD4* overexpression was also reported to correlate with poor survival and was significantly higher in platinum-resistant HGSC [89]. 

Finally, ISWI family members also display genetic status abnormalities in human cancers, including OCs [268]. Interestingly, deregulated expression is closely linked to drug response and patient outcome. 

### 4.2. Checkpoint Factors

DNA damage and replication stress initiate the DDR through the activity of two signaling kinase proteins: ATM and ATR, both belonging to the phosphatidylinositol-3 (PI-3) kinase-like kinase family (PIKK). ATM is generally activated by a DSB, whereas ATR is activated by a SSB, DNA replication stress, and DNA-end resection, which occurs during DSB repair, as mentioned before [48]. Upon their recruitment to DNA damage sites, both kinases activate the DNA damage checkpoints, which arrest the cell cycle allowing time for repair. The response is performed through the phosphorylation and activation of the Checkpoint Kinase 2 (CHK2), by ATM, and Checkpoint kinase 1 (CHK1), by ATR, although an extensive communication exists between the two signaling pathways. 

One of the main substrates of CHK2 is p53, whose activation by phosphorylation promotes the upregulating of p21, an inhibitor of cyclin-dependent kinases that induce a G1/S transition arrest [48,49]. Activation of CHK2 also phosphorylates CDC25C leading to its degradation, which prevents the activation of downstream signaling pathways such as p21 and cell cycle B1 and result in a G2/M arrest [49].

CHK1, activated by ATR, inhibits S phase DNA replication and G2/M phase transition through the phosphorylation and activation of WEE1 and CDC25C [50]. The kinase WEE1 regulates the entry into mitosis by negatively controlling the cyclin-dependent kinases CDK1 and CDK2. 

Germinal mutations in *ATM* produce Ataxia-telangiectasia, an autosomal recessive neurodegenerative disorder with an increased risk of developing cancer (40%), particularly leukemias and lymphomas [269]. Somatic *ATM* mutations also occur in several sporadic tumor types, especially in hematologic malignancies [91]. The risk of OC has been found to be slightly elevated for people with an inherited *ATM* mutation (a lifetime risk of about 2–3% versus 1.3% for the general population) [92], and the percentage of OC tumors with *ATM* somatic mutations seems to depend on the OC subtype. A recent study analyzing 207 ovarian cancer samples from a Japanese population reported that *ATM* mutations are more frequent in CCC (9%) and EC patients (18%) than in HGSC patients (4%) [270].

Mutations in *ATR* are much rarer that in *ATM*. They are not associated with hereditary breast and ovarian cancer (HBOC) syndrome, only with Seckel syndrome, an autosomal recessive disorder not implicated in malignancy [271]. 

Regarding *CHK2*, genetic testing for various hereditary cancer predispositions has identified mutations in this gene among the most frequent germline alterations. However, despite many published results, the association of *CHK2* mutations with OC can be neither confirmed nor rejected, due to the presence of many variants of unknown significance (VUS) that affect clinical interpretation [272]. On the other hand, somatic mutations of *CHK2* have been reported in small subsets of diverse types of sporadic cancers including OC [93]. On the contrary, and despite playing a central function in the DDR, no germline, or somatic mutations in *CHK1* have been conclusively associated with human disease [273], which is probably due to its essential role in cell proliferation and survival. 

#### 4.2.1. ATM Inhibitors

ATM is considered a tumor suppressor. *ATM* mutations are predicted to result in enhanced sensitivity to platinum chemotherapy [91]. However, when ATM is present in tumor cells, it confers resistance to ionizing radiation and DNA-damaging agents. For this reason, ATM inhibitors have been developed for use in cancer therapy and have been reported less harmful for non-tumoral cells [274]. 

Multiple preclinical studies have analyzed the efficacy of ATM inhibitors in monotherapy or in combination with other chemotherapeutic drugs in several tumors, including OCs (Table 4). In general, these inhibitors decreased OC cell proliferation and synergized with other compounds, such as fenofibrate, an inhibitor of PPARA, 673A, or DNA-damaging agents, including ionizing radiation, trabectedin, and lurbinectedin [123,124,125,126,275]. 

#### 4.2.2. ATR Inhibitors

Like ATM, ATR inhibition decreases both DNA checkpoint response and DSB repair, enhancing the efficacy of IR and DNA-damaging drugs. Moreover, it is known that p53- or ATM-defective cells can only rely on ATR to avoid a mitotic catastrophe for excessive DNA damage accumulation after these treatments. Based on this knowledge, many ATR inhibitors (ATRi) are under preclinical and clinical investigation as monotherapies or in combination with other anticancer agents such as cisplatin, topotecan, gemcitabine, trabectedin, or PARPi [125,127,128,276,277,278,279]. Preclinical results have shown inhibition of cell proliferation [127,129,136,137,139] and, in many cases, synergistic effects in combination with different drugs [124,125,128,130,131,132,133,134,135,138,140,141,142,145,279], which are summarized in Table 4. Clinical results have shown that ATR inhibitors (celarasertib, berzosertib, elimusertib, and gartisertib) were safe and well tolerated and presented preliminary antitumor activity in OC patients [201,280]. In addition, it has also been studied its combination with other chemotherapeutic drugs such as PARP inhibitors [198,199], carboplatin [200], cisplatin [199,201,202], topotecan [200,203], or gemcitabine [204,281]. Clinical results of ATR inhibitors are summarized in Table 5. 

#### 4.2.3. CHK1 and CHK2 Inhibition

Several CHK1and CHK2 inhibitors with different potency and selectivity have been developed. Numerous CHK1 inhibitors have been studied in preclinical studies such as SRA737, V158411, LY2880070, MK-8776, or PF-477736. These inhibitors caused inhibition of cell proliferation of OC cells and enhanced sensitivity to DNA-damaging agents [128,133,144,146]. The inhibitor SRA737 was reported to synergistically enhance the cytotoxicity of PARPi (niraparib, olaparib) in OC cells [143]. Moreover, treatment with SRA737 in HGSC patients-derived xenograft models, where PARPi showed limited activity, resulted in a significant stabilization of the disease [282]. A phase I/II trial has tested the security and efficacy of SRA737 in monotherapy or in combination with gemcitabine and cisplatin in several cancer patients, including HGSC OC patients [205,206]. It was reported that low doses of gemcitabine could increase their activity by induction of replication stress [206]. Moreover, LY2880070 has also been tested in clinical trials together with low doses of gemcitabine and has proven to be well-tolerated in OC patients [207].

Regarding CHK2 inhibitors, PV1019, PHI-101, C342, and AZD7762 have shown to inhibit OC cell proliferation and synergistically increased the cytotoxic effect of DNA-damaging agents [147,148,149,150,151]. PHI-101, which elicits a synergistic lethal response in combination with olaparib regardless of *BRCA* and *TP53* status, potentiated the toxicity triggered by genotoxic agents such as cisplatin and topotecan. Recently, a phase I clinical trial has started and will evaluate the safety and tolerability of PHI-101 in platinum-resistant recurrent OC patients [283] (Table 5).

Finally, CHK1/2 dual inhibitors, such as prexasertib, have also been developed. Prexasertib (KY2606368 or LY2606368) has shown antitumor activity in HGSC OC cells and HGSC OC patient-derived xenografts. Moreover, its combination with olaparib or gemcitabine induced a synergistic cytotoxic effect [152,153,284]. Several clinical trials are studying the safety and tolerability of prexasertib on OC patients, in monotherapy or in combination with other chemotherapeutic drugs. It has been described that prexasertib treatment was safe and well tolerated and presented preliminary antitumor activity in OC patients [208,209,285]. Its combination with olaparib showed clinical activity in patients that had previously progressed after PARPi treatment [210].

#### 4.2.4. WEE1 Inhibition

Inhibitors of WEE1, a negative regulator of entry into mitosis, have also been developed. Adavosertib (AZD1775 or MK1775), has shown promising results against several tumor cell lines, including OC. In OC cell lines, adavosertib exerted an antitumor activity by inhibiting cell proliferation and migration and inducing DNA damage, apoptosis, and G2/M cell cycle arrest. Moreover, adavosertib activity seemed to be independent of HR repair status [154,286,287]. Patient-derived organoids (PDO) studies showed that this compound could be useful for the treatment of *TP53*-mutated OC patients [287]. In addition, several studies have proven that adavosertib in combination with other drugs, such as AZD6738 (ATR inhibitor), PF-00477736 (CHK1inhibitor), or radioimmunotherapy, enhanced cytotoxicity obtaining a synergistic cytotoxic effect with combined treatments [155,156,157]. Currently, several clinical trials (phase I and II) are testing the efficacy of adavosertib in combination with other chemotherapeutic drugs, such as gemcitabine, paclitaxel, carboplatin, olaparib, or PLD in OC patients. Adavosertib treatment presented manageable toxicity profiles and its combination with gemcitabine [213], carboplatin with/without paclitaxel [211,212,214], and oaparib [215] could benefit OC patients as summarized in Table 5.

### 4.3. p53 Pathway

The protein p53 exerts an essential role in the maintenance of genome integrity; it activates some DNA repair proteins when DNA has been damaged, arrests the cell cycle at the G1/S transition allowing DNA repair, and induces apoptosis when DNA damage proves to be irreparable [288]. Because of these essential roles in tumor suppression, p53 is unsurprisingly found mutated in many cancers. In fact, more than 50% of all types of human cancers bear a *TP53* mutation [94]. These mutations are especially prevalent in the OC subtype HGSC since they have been identified in 96% of cases [94,95]. Missense mutations in the regions encoding the DNA-binding domains of p53 are the most frequent. They appear in early stages of the disease and are considered driver mutations in ovarian carcinogenesis that can be followed by deletions or loss of heterozygosity (LOH) of chromosomes carrying *TP53*, *BRCA1*, or *BRCA2* [96]. 

Several therapeutic strategies have been designed to increase or restore the p53 response in human cancers [289,290]. The most promising are those that try to restore the tumor suppressor protein in cells carrying *TP53* gene mutations. PRIMA-1 (also known as APR-017) and its methylated analog PRIMA-1^MET^ (APR-246) are low molecular weight compounds that induce apoptosis in tumor cells by restoring the transcriptional function of mutant p53 [158,291]. PRIMA-1 has shown to induce cell death of OC cells, especially those with mutant p53, and re-sensitized chemoresistant-OC cells with mutant *TP53* to cisplatin [158,159]. PRIMA-1^MET^ is a prodrug, which is converted to the active compound methylene quinuclidinone (MQ), that binds to cysteine residues in mutant p53 and restores its wild-type conformation [160]. This agent re-sensitized cisplatin-resistant or doxorubicin-resistant OC cells to cisplatin and doxorubicin, respectively, in vitro and in vivo [160]. In addition, PRIMA-1^MET^, together with cisplatin/carboplatin/doxorubicin/gemcitabine, exerted a synergistic cytotoxic effect in OC cells [160] that was also observed in primary cancer cells isolated from HGSC OC patients with missense *TP53* mutations [160,161]. Its security and effectivity together with PLD/carboplatin have been explored in a phase 1b study in relapsed platinum-sensitive HGSC OC patients [218] (Table 5). Another agent able to restore the p53 function is ReACP53 [162]. This peptide can penetrate tumor cells and inhibit p53 amyloid formation and aggregation, which might rescue p53 function [162,163]. Its preclinical effect in OC cells is detailed in Table 4. 

Another way to increase p53 activity is by the inhibition of its negative regulators MDM2/MDM4, that are often overexpressed in tumors WT for *TP53*. The first molecule identified as a potent inhibitor of the p53-MDM2 interaction was nutlin [292]. The efficacy of this molecule requires WT status of the TP53 gene and intact p53 signaling machinery. In OC, most HGSC tumors harbor mutations of this gene, as previously mentioned; however, CCC or LGSC usually express WT p53 [164]. In vitro experiments in OC *TP53* WT cells, found that nutlin reduced cell viability and induced apoptosis [164]. Moreover, it exerted a synergistic cytotoxic effect together with other DNA-damaging drugs and PARPi [165,166,167,168,169,293]. A similar effect was described for other MDM2 inhibitors, such as RG7388 [167,169] and RG7112, which reduced the growth of clear cell tumor cells with intact TP53 both in vitro and in vivo [170]. 

In summary, we have reviewed the main DDR pathways involved in the maintenance of genome stability; the core DNA repair pathways (DR, MMR, NER, BER, HR, and NHEJ); and the complementary processes that also contribute to overall genome maintenance: cell cycle checkpoints, the p53 pathway, and chromatin remodeling factors. All of them have been found to be altered in OCs either through pathogenic mutations, epigenetic alterations, or polymorphisms in DDR genes. These alterations may contribute to the onset of the disease but also affect sensitivity to therapy. Therefore, considerable efforts are being dedicated to target these defects.

## 5. Conclusions and Future Perspectives

Alterations in the DDR are commonly observed in OC. The most frequent is HR deficiency (HRD), which has been detected in approximately 50% of epithelial OCs. In 10% of the HGSC subtype, the HRD is caused by germline mutations in the *BRCA1/2* genes. These tumors depend on PARP-mediated base excision repair for survival and are selectively killed by PARPi, such as olaparib, that has been approved by the FDA and the EMA for recurrent epithelial OC. In addition to *BRCA1* and *BRCA2*, mutations in other HR genes have been detected in OCs and confer an HR deficiency known as “BRCAness” status. Patients carrying these tumors are also predicted to benefit from the synthetic lethal approach using PARPi. Consequently, several HRD assays have been developed to identify and stratify the patients, although they still have several limitations that need to be solved.

Many different studies, summarized in this review, have also detected other DDR alterations in a variable percentage of OCs, such as defects in DR, MMR, NER, and NHEJ repair pathways, or alterations in chromatin remodelers, checkpoint proteins, and the p53 pathway. Two important elements are needed to specifically target these tumors: biomarkers or reliable functional assays to determine the specific DDR defect in OC samples, and targeted therapies to the associated vulnerabilities. In some cases, laboratory methods could identify the deficiencies (DNA sequencing, determination of MMR or p53 status), but the strategies to specifically target those tumors are not yet available. 

Nevertheless, systematic next generation sequencing of individual tumors, or the use or DDR gene panels, will help to not only identify clinically actionable mutations, but also guide patient selection for new clinical studies. In this regard, increasing evidence suggests that cancers with DDR mutations may have high mutational loads and neo-antigens. Therefore, the combination of DDR inhibitors with immunotherapy appears promising to fight these tumors. Defects in the DDR might also be induced by gene silencing through epigenetic mechanisms or by dysregulation of gene function. Reliable functional assays to identify these defects in tumor samples needs to be developed to increase the number of patients that might also respond to DNA damage/repair targeted therapeutic drugs. In addition, more investigation is needed to identify new vulnerabilities associated with specific defects in the DDR and with the acquisition of resistance. These vulnerabilities could be translated into new therapeutic strategies.

Targeting DDR signaling pathways has also become an attractive strategy for increasing the effect of DNA-damaging drugs and overcoming tumor resistance. The idea is to find drug combinations that work in an additive, or better, in a synergistic manner, that is, when the effect of two or more agents working in combination is greater than the expected additive effect. These approaches deserve more investigation since they increase the potential to overcome drug resistance and allow a lower therapeutic dosage of each individual drug, which reduces toxicity. A large number of DDR inhibitors with different potency and selectivity have been developed and are being tested in preclinical and clinical trials. Many of them decrease OC cell proliferation and show synergistic cytotoxic effects in combination with different genotoxic drugs. The challenge ahead is to translate these basic studies into clinical applications that increase the therapeutic arsenal to fight the disease.

## Figures and Tables

**Figure 1 cancers-15-00448-f001:**
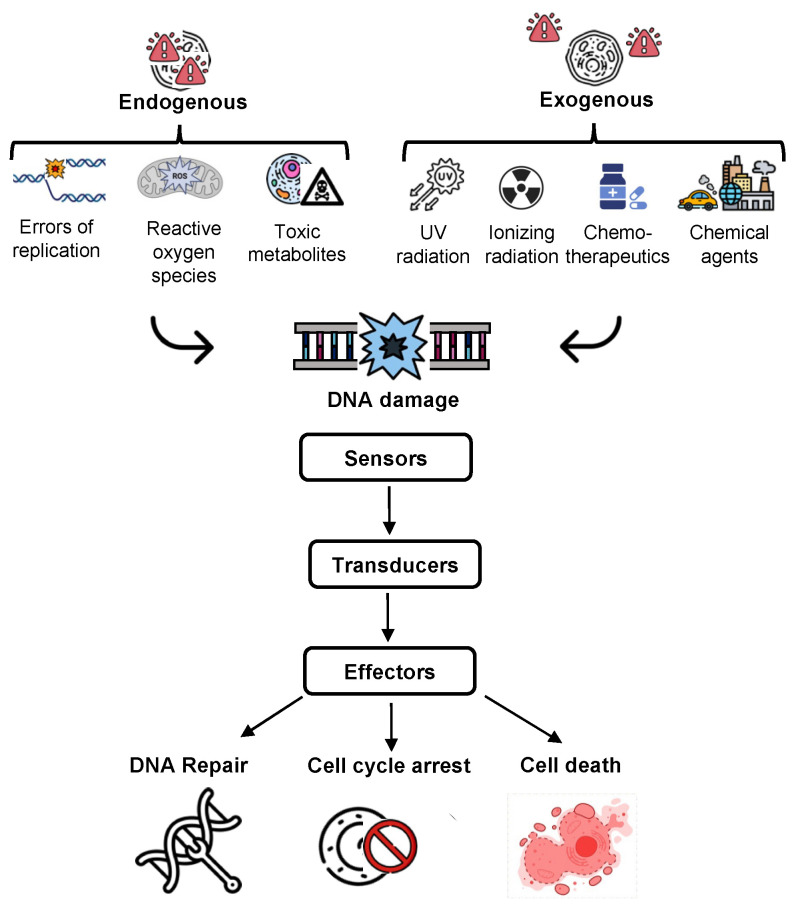
Simplified representation of the DNA damage response (DDR) activated by endogenous or exogenous DNA-damaging agents. This figure has been created using images from BioRender.com and Flaticon.com.

**Figure 2 cancers-15-00448-f002:**
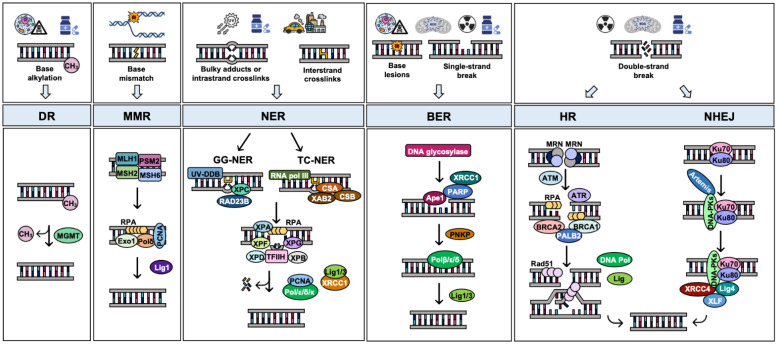
Schematic representation of the five major DNA repair pathways and the proteins involved. This figure has been created using images from BioRender.com and Flaticon.com.

**Table 1 cancers-15-00448-t001:** Clinical characteristics and genetic alterations in the five different subtypes of epithelial OC [2,5].

Subtype	High-Grade Serous	Endometrioid	Clear cell	Mucinous	Low-Grade Serous
Prevalence	70%	10%	10%	5%	<5%
Stage at diagnosis	Advanced	Early	Early	Early	Early or advanced
Progression speed	High	Low (90%) High (10%)	Low	Low (50%) High (50%)	Low
Genetic alterations	*TP53* *BRCA1* *BRCA2*	*CTNNB1* *PIK3CA* *ARID1A* *KRAS* *PPP2R1* *PTEN*	*ARID1A* *PTEN* *PIK3CA* *KRAS* *MET*	*KRAS* *TP53* *HER2/Neu*	*BRAF* *KRAS* *HER2/Neu*
Chemotherapeutic response	High (at early stages)	Low	Low	Low	Low

**Table 2 cancers-15-00448-t002:** Main proteins involved in each DDR pathway.

DDR Pathway		Proteins Involved (References)		
DR		MGMT [28]		
MMR		Sensors	MSH2, MSH6, MLH1, PMS1, PMS2, and MSH3 [29,30,31]	
		Transducers	RPA, PCNA, and RFC [32]	
		Effectors	Exo1, Pol δ, and DNA ligase I [32,33]	
NER		Sensors	TC-NER	RNA polymerase, CSA, CSB, and XAB2 [26,34]
			GG-NER	XPC-RAD23B and UV-DDB [26,34]
		Transducers	XPA, XPG, XPF/ERCC1, RPA, TFIIH complex [34,35]	
		Effectors	PCNA, RFC, DNA polymerases δ, ε, and κ, and DNA ligase I and LIG3 [34,35]	
BER		Sensors	DNA glycosylase (OGG1, UNG, or MUTYH), APE1, and PARP1 [36]	
		Transducers	PNKP [37,38]	
		Effectors	XRCC1, LIG3, DNA polymerase β [37,38]	
HR		Sensors	MRN complex (MRE11, RAD50, NSB1) [39,40]	
		Transducers	ATM, BRCA1, ATR, RPA, PALB2, BRCA2 [41,42]	
		Effectors	RAD51, DNA ligase, DNA polymerase [39,40]	
NHEJ		Sensors	C-NHEJ	Ku70, Ku80 [43,44]
			Alt-NHEJ	PARP1 [45,46]
		Transducers	DNA-PKcs, ATM, ATR, and Artemis [43,44]	
		Effectors	C-NHEJ	XRCC4, XLF, and DNA ligase IV [43,44]
			Alt-NHEJ	XRCC1, DNA polymerase Θ, and LIG3 [45,46]
Associated pathways	Chromatin remodelers	SWI/SNF, INO80, CHD, ISWI [47]		
	Checkpoints factors	Transducers	ATM, ATR, CHK1, CHK2 [48]	
		Effectors	p53, p21, CDC25C, WEE1, CDK1, CDK2 [48,49,50]	

**Table 3 cancers-15-00448-t003:** DDR alterations in OCs.

DDR Pathway	Genomic/Epigenomic Alterations in OC (References)
DR	MGMT promoter hypermethylation [51,52]
MMR	Mutations in *MLH1*, *MSH2*, *MSH6,* and *PMS2* [53,54,55] *MLH1* promoter hypermethylation [56,57,58]
NER	SNPs in NER genes [59] Homozygous deletions, missense, or splice site mutations in NER genes [60,61]
BER	SNPs in *OGG1*, *APE1*, and *XRCC1* [62,63,64,65,66,67,68] *APE1* overexpression [69]
HR	Genetic and epigenetic modifications of genes encoding HR proteins [70,71,72] Mutations in *BRCA1, BRCA2, RAD51C, RAD51D* and MRN complex genes [70,71,72,73,74,75,76] Downregulation of *RAD50* [75,77] *RAD51* promoter hypermethylation [60]
NHEJ	Mutations or overexpression of genes that encoded for NHEJ proteins (DNA-PKs, DNA polymerase Θ, or XRCC4) [60,78,79,80] SNPs in NHEJ genes (DNA ligase IV, *XRCC1*) [60]
Chromatin remodelers	Mutations in *ARID1A* [81,82,83,84] Mutations in *CHD4* [85,86,87], *CHD5* [88], and *CHD8* [88] subunits Amplification of CHD4 [85,89] and ACT6LA [88,90] subunits
Checkpoints factors	Mutations in *ATM* [91,92]
	Somatic mutations in *CHK2* [93]
p53 pathway	Mutations in *TP53* [94,95] Loss of heterozygosity in the chromosome that contains *TP53* [96]

**Table 4 cancers-15-00448-t004:** DDR-targeting drugs investigated in preclinical studies.

DDR Pathway	Drug	Target(s)	Preclinical Evidence(s) in OC Cells or Xenograft Models
DR	**PaTrin-2**	MGMT	Sensitization to temozolomide [97]
NER	MCI13E	RPA	Antitumor activity [98]
	TDRL-505	RPA	Antitumor activity [98]
	TDRL-551	RPA	Antitumor activity and synergistic cytotoxic effect in combination with etoposide and platinum [98]
	**Trabectedin/** **Lurbinectedin**	NER and HR proteins	Antitumor activity and synergistic cytotoxic effect in combination with cisplatin, doxorubicin, and irinotecan [99,100,101,102]
BER	**Methoxyamine**	APE1	Enhancement of temozolomide cytotoxicity [103]
	E3330 and analogs	APE1	Inhibition of cell proliferation [104,105]
	Spiclomazine/ fiduxosin	APE1/NPM1 interaction	Inhibition of cell proliferation and sensitization to bleomycin [106]
	**PARPi**	PARP	Specific killing of *BRCA*-deficient tumors [107,108]
HR	Mirin	Mre11	Increase sensitivity to DNA-damaging agents (cisplatin, carboplatin, chloroquine) [109,110,111]
	**Panobinostat**	Rad51	Inhibition of OC cell proliferation, induction of apoptosis, and inhibition of DNA repair by altering the correct repair of Rad51 [109,112,113]. Synergistic cytotoxic effect in combination with chloroquine or cisplatin [109,114]
	Ellagic acid/ luteolin	MRN complex	Decrease cellular proliferation and migration [115]
NHEJ	NU-7026/NU-7441	DNA-PKs	Enhancement of DNA-damaging agents’ cytotoxicity (irradiation, chloroquine, cisplatin) [116,117,118]
	**Peposertib**	DNA-PKs	Increase radiosensitivity and chemosensitivity to etoposide or doxorubicin [119,120]
	AZD7648	DNA-PKs	Increase the cytotoxicity to irradiation, doxorubicin, or PARP inhibitors [121,122]
Checkpoints factors	KU-60019	ATM	Inhibition of cell migration and induction of apoptosis [123]. Synergistic cytotoxic effect in combination with fenofibrate (PPARA inhibitor) [123]. Sensitization to ionizing radiation, trabectedin, and lurbinectedin [124,125]
	AZD0156	ATM	Synergistic cytotoxic effect in combination with fenofibrate (PPARA inhibitor) [123]
	KU-55933	ATM	Sensitization to ionizing radiation, trabectedin and lurbinectedin [124,125]
	AZD1390	ATM	Synergistic cytotoxic effect in combination with an aldehyde dehydrogenase 1 inhibitor (67A) [126].
	VE-821	ATR	Inhibit cell proliferation and enhancement of cisplatin, topotecan, gemcitabine, and veliparib cytotoxicity [127,128,129,130,131]. Enhancement of lurbinectedin and trabectedin cytotoxicity in combination with KU-60019 [125]
	AZ20	ATR	Sensitize PARPi-resistant cells to PARPi [131]
	**Ceralasertib**	ATR	Synergistic cytotoxic effect in combination with belotecan, an aldehyde dehydrogenase 1 inhibitor (67A), and PARPi [126,132,133,134,135]
	**Berzosertib**	ATR	Reduction of cell proliferation and cell survival [127,129,136,137]
	EPT-46464	ATR	Synergistic cytotoxic effect in combination with cisplatin, carboplatin, and radiation [124]
	NU6027	ATR/ CDK2	Enhancement of the cytotoxic effect of cisplatin and temozolomide [138]
	**Elimusertib**	ATR	Inhibition of cell proliferation [139,140] and enhancement of the cytotoxic effect of carboplatin and therapeutic radiopharmaceuticals [141,140]
	**Gartisertib**	ATR	Enhancement of topotecan, irinotecan, gemcitabine, cisplatin, and talazoparib cytotoxicity [142]
	**SRA737**	CHK1	Synergistic cytotoxic effect in combination with PARPi [143]
	V158411	CHK1	Inhibition of cell proliferation and enhancement of carboplatin and cisplatin cytotoxicity [144]
	MK-8776	CHK1	Inhibition of cell proliferation and enhancement of gemcitabine and olaparib efficacy [128,133,145]
	PF-477736	CHK1	Synergistic antiproliferative effect in combination with topotecan [146]
	PV1019	CHK2	Inhibition of cell proliferation and synergistic cytotoxic effect in combination with topotecan or camptothecins [147]
	**PHI-101**	CHK2	Antitumor activity [148]
	C3742	CHK2	Synergistic cytotoxic effect in combination with cisplatin [149]
	AZD7762	CHK2	Synergistic cytotoxic effect in combination with cisplatin [150]. Sensitization to PARG inhibition [151]
	**Prexasertib**	CHK1/ CHK2	Antitumor activity and synergistic cytotoxic effect in combination with olaparib or gemcitabine [128,152,153]
	**Adavosertib**	WEE1	Antitumor activity, and inhibition of cell proliferation and migration. Induction of DNA damage, apoptosis, and G2/M cell cycle arrest [154]. Synergistic cytotoxic effect in combination with ATR inhibitor (AZD6738) [155], CHK1inhibitor (PF-00477736) [156], and radioimmunotherapy [157]
p53 pathway	PRIMA-1	Mutated p53	Induction of cell death and re-sensitization of chemoresistant-cells to cisplatin [158,159]
	**PRIMA-1^MET^**	Mutated p53	Re-sensitization of cisplatin/doxorubicin-cells to cisplatin/doxorubicin [160]. Synergistic cytotoxic effect in combination with cisplatin, carboplatin, or doxorubicin [160,161]
	ReACP53	Mutated p53	Cell proliferation decrease and synergistic cytotoxic effect in combination with carboplatin [162,163]
	Nutlin	MDM2	Reduction of cell viability, induction of apoptosis, and synergistic cytotoxic effect in combination with cisplatin, rucaparib, or etoposide [164,165,166,167,168]
	RG7388	MDM2	Reduction of cell viability, induction of apoptosis, and synergistic cytotoxic effect in combination with cisplatin, rucaparib, or etoposide [164,167,169]
	RG7112	MDM2	Cell growth reduction [170]

Drugs in bold have been tested in clinical trials.

**Table 5 cancers-15-00448-t005:** DDR-targeting drugs investigated in clinical studies in monotherapy or in combination with other antitumor drugs.

DDR Pathway	Target(s)	Drug	Combined with	Condition	Phase	Clinical ID	Results
DR	MGMT	PaTrin-2	Temozolomide	OC	I	[171]	One patient with OC has a decrease of around 50% of tumor markers and stable radiology over 5–6 cycles of treatment [171]
NER	NER and HR proteins	Trabectedin	-	Advanced OC	II	NCT00050414	Trabectedin was active and well-tolerated in advanced OC platinum-sensitive patients. The optimal regimen was established [172]
				Advanced tumor malignancies	II	NCT00786838	Trabectedin efficacy was confirmed [173]
				Advanced soft tissue sarcomas	II	NCT00003939	Trabectedin has proven to control tumor progression in highly pretreated, progression, advanced, metastatic resistant, or refractory sarcoma patients [174]
				Ovarian carcinosarcoma	II	NCT02993705	Trabectedin conferred a modest benefit to patients with advanced OC and it was well-tolerated [175]
				*BRCA* mutated OC	II III	NCT01772979 NCT02903004	OC patients with BRCAness phenotype could benefit from trabectedin treatment. However, this treatment did not improve survival compared to standard chemotherapy in *BRCA-*mutated and BRCAness phenotype OC patients [176,177]
			PLD	Relapsed OC	II	NCT04887961	No results posted
				Partially platinum-sensitive OC	III	NCT01379989	Trabectedin/PLD combination showed a similar overall survival that carboplatin/PLD combination and could be considered for treating patients who need a longer recovery time from platinum toxicities [178]
				Advanced relapsed OC	III	NCT00113607 NCT01846611	Trabectedin/PLD combination did not show a favorable overall survival benefit or safety. Specifically, patients with *BRCA*-mutation or a platinum-free interval of 6–12 months seemed to present a survival benefit from this combination [179,180,181,182]
				Recurrent OC	III	NCT03690739	No results posted
				Platinum-sensitive recurrent OC	IV	NCT03164980	OC patients treated with trabectedin/PLD did not show inferiority signals compared to standard platinum-based chemotherapy. The study is ongoing and recruiting new patients [183]
				Platinum-sensitive recurrent OC	Observational	NCT02394015 NCT05512676	The intercalation with a nonplatinum regimen, such as trabectedin/PLD, could improve the response to a platinum-base therapy platinum-sensitive OC patients [184]
				Partially platinum-sensitive recurrent OC	Observational	NCT03446495	No results posted
				Relapsed OC	Observational	NCT02825420	The combination of trabectedin and PLD represented a therapeutical safe option for platinum-sensitive recurrent OC regardless of prior anti-angiogenic treatment [185]
				Platinum-sensitive relapsed OC	Observational	NCT02163720 NCT01869400	PLD/trabectedin supposed a therapeutic option for partially or fully platinum-sensitive recurrent OC patients [186,187]
			PLD +/− olaparib	Recurrent OC	II	NCT03470805	No results posted
			Durvalumab	OC	I	NCT03085225	The combination of trabectedin and durvalumab presented a manageable toxicity and a promising antitumor activity in platinum-refractory OC patients [188]
			Docetaxel + pegfilgrastim/filgastrim	Recurrent or persistent OC	II	NCT00569673	Docetaxel/trabectedin was well-tolerated and seemed to be more active than docetaxel treatment in recurrent OC patients [189]
			Bevacizumab +/− carboplatin	Recurrent OC	II	NCT01735071	Bevacizumab/trabectedin combination had clinical activity and could be a therapeutic option for partially platinum-sensitive OC patients. Bevacizumab/trabectedin/carboplatin demonstrated a positive activity and should be further studied [190]
		Lurbinectedin	Paclitaxel +/− bevacizumab	Advanced solid tumors	I	NCT01831089	Lubinectedin combined with paclitaxel and/or bevacizumab presented a manageable toxicity and promising antitumor activity in patients with advanced solid tumors, including OC [191]
			Olaparib	Advanced solid tumors	I/II	NCT02684318	Lurbinectedin/olaparib combination was feasible and recommended doses were obtained for each drug [192]
			Irinotecan	Solid tumors	I/II	NCT02611024	One OC *BRCA1*-mutated patient presented an extraordinary response with a time to further progression of 8 months. No more results have been posted [193]
BER	APE1	Methoxyamine	-	Platinum-resistant OC	III	NCT02421588	Lurbinectedin showed antitumor activity similar to PLD and it was better tolerated [194]
			Temozolomide	Granulosa cell OC	I/II	NCT01851369	Two patients with granulosa cell OC experienced a partial response [195]
			Permetrexed + cisplatin	Advanced solid tumors	I/II	NCT02535312	No results posted
	PARP	Olaparib	Described in Appendix A				
		Niraparib					
		Talazoparib					
		Pamiparib					
		Rucaparib					
HR	Rad51	Panobinostat	-	Advanced solid tumors	I	NCT00739414	Panobinostat treatment was safe and potentially effective against advanced solid tumors like OC [196]
			Gemcitabine	Solid tumors	I	NCT00550199	Recommended doses for panobinostat/gemcitabine were established [197]
NHEJ	DNA-PKs	Peposertib	PLD	Recurrent OC	I	NCT04092270	No results posted
Checkpoints factors	ATR	Ceralasertib	Olaparib	Recurrent OC	II	NCT03462342	The combination of ceralasertib and olaparib was well-tolerated. No objective response was observed; however, a signal of activity was observed and depended on *BRCA1* status [198]
				Recurrent OC	II	NCT03579316	No results posted about ceralasertib/olaparib combination
				Gynecological cancers	II	NCT04065269	No results posted
				Advanced solid tumors	II	NCT02576444	Ceralasertib/olaparib combination has demonstrated preliminary activity in *ATM-*mutated tumors and in *BRCA*-mutated PARPi-resistant HGSC OC patients [199]
			Carboplatin/Olaparib/Durvalumab	Advanced tumors	I	NCT02264678	No results posted
		Berzosertib	Carboplatin + Gemcitabine	Recurrent and metastatic OC	I	NCT02627443	No results posted
			Carboplatin + Avelumab	PARPi resistant, recurrent, and platinum sensitive OC	I	NCT03704467	Berzosertib/carboplatin/carboplatin safe doses were established; however, phase II was not started [200]
			Gemcitabine/Cisplatin/Etoposide/Carboplatin/Irinotecan	Advanced solid tumors	I	NCT02157792	Berzosertib/cisplatin and berzosertib/carboplatin combinations were well-tolerated and presented preliminary preclinical activity in patients with advanced solid tumors, including OC patients. [201,202]
			Topotecan	Ovarian neoplasms	I	NCT02487095	Only one patient with OC was recruited, this patient presented a short duration of the response and a progressive disease [203]
			Gemcitabine	Recurrent OC	II	NCT02595892	The addition of berzosertib to gemcitabine in platinum-resistant HGSC increased PFS rate compared to gemcitabine in monotherapy [204]
		Gartisertib	Niraparib	PARPi resistant and recurrent OC	I	NCT04149145	Not yet recruiting patients
		Elimusertib	Niraparib	Advanced OC	I	NCT04267939	No results posted
			Gemcitabine	Advanced OC	I	NCT04616534	No results posted
			Gemcitabine +/− Cisplatin	Advanced OC	I	NCT04491942	No results posted
	CHK1	SRA737	-	HGSC OC with/without *CCNE1* gene amplification	I/II	NCT02797964	SRA737-maximum tolerated dose was established. Based on tolerability and pharmacokinetics, phase II was recommended [205]
			Gemcitabine +/− Cisplatin	HGSC OC	I/II	NCT02797977	Low-dose gemcitabine combined with SRA737 has been well-tolerated in HGSC OC patients [206]
		LY2880070	+/− Gemcitabine	Advanced or metastatic OC	I/II	NCT02632448	LY2880070 together with low-dose gemcitabine was well-tolerated [207]
	CHK2	PHI-101	-	Platinum-resistant or refractory OC	I	NCT04678102	No results posted
	CHK1/ CHK2	Prexasertib	-	Platinum-resistant or refractory OC	II	NCT03414047	Prexasertib has demonstrated durable single agent activity in a subset of OC patients regardless their clinical characteristics, *BRCA* status of prior therapies [208]
			-	HGSC OC with/without *BRCA* mutations	II	NCT02203513	Prexasertib presented clinical activity and was tolerable in HGSC OC patients with *BRCA-*wild type [209]
			Olaparib	Advanced solid tumors	I	NCT03057145	The combination of prexasertib and olaparib had preclinical activity in *BRCA*-mutant HGSC OC patients who had previously progressed on a PARPi [210]
	WEE1	Adavosertib	-	Advanced OC	I	NCT02659241	No results posted
			Carboplatin/Paclitaxel/Gemcitabine/PLD	Platinum-resistant OC	II	NCT02272790	3% of platinum-resistant OC patients presented a completed response and 29% a partial response. The highest response rate was obtained with adavosertib/carboplatin combination [211]
			Carboplatin	*TP53-*mutated refractory and resistant OC	II	NCT01164995	Adavosertib/carboplatin combination demonstrated a manageable toxicity. The overall response rate was 43% and one patient (5%) presented prolonged complete response [212]
			Gemcitabine	Recurrent OC	II	NCT02101775	Recurrent OC patients-treated with adavosertib/gemcitabine presented a longer PFS [213]
			Paclitaxel + carboplatin	*TP53*-mutated platinum- sensitive OC	II	NCT01357161	The addition of adavosertib to chemotherapy treatment (paclitaxel/carboplatin) improved PFS [214]
			Olaparib	Recurrent OC Refractory solid tumors Advanced solid tumors	II I II	NCT03579316 NCT02511795 NCT02576444	Adavosertib in monotherapy or combined with olaparib demonstrated efficacy in patients with resistance to PARPi. Adavosertib/olaparib combination presented manageable toxicities [215,216,217]
		ZN-c3	Niraparib	Platinum-resistant OC	I/II	NCT05198804	No results posted, recruiting patients
p53 pathway	Mutated p53	PRIMA-1^MET^	PLD	Platinum-resistant HGSC OC	II	NCT03268382	36 patients were enrolled in this study which have been treated with several doses of PLD and APR-246. That combination was feasible and adverse effects were manageable
			+/− Carboplatin/PLD	Recurrent HGSC OC	I/II	NCT02098343	APR-246/carboplatin or APR-246/PLD combinations were effective in HGSC OC patients with *TP53-*mutated and recommended phase II dose has been established [218]

HGSC: high-grade serous carcinoma; PARPi: PARP inhibitor; PLD: pegylated liposomal doxorubicin; PFS: progression-free survival; ORR: objective response ratio; OS: overall survival.

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
