# Peer review of "DNA Damage Response Alterations in Ovarian Cancer: From Molecular Mechanisms to Therapeutic Opportunities"

_cancers, 2023, doi:10.3390/cancers15020448_

Round 1

Reviewer 1 Report

The manuscript represents a comprehensive review of pathogenic nucleotide variants in ovarian cancer (OC) and therapy, related to these variants. 

The authors observed clinical characteristics and molecular mechanisms of pathogenicity development for different OC subtypes.

Besides, the authors summarized the main conclusion in informative tables and figures.

The references of the current manuscript contain 13% of the reviews, the most cited journal was Gynecological Oncology, the median year of publication is 2017, and the most frequent words from titles of references reflect the topic of the current work (see analysis_of_references.pdf). 

My suggestion for authors is to describe the way they searched and selected literature for the review.

Author Response

The manuscript represents a comprehensive review of pathogenic nucleotide variants in ovarian cancer (OC) and therapy, related to these variants. The authors observed clinical characteristics and molecular mechanisms of pathogenicity development for different OC subtypes. Besides, the authors summarized the main conclusion in informative tables and figures. The references of the current manuscript contain 13% of the reviews, the most cited journal was Gynecological Oncology, the median year of publication is 2017, and the most frequent words from titles of references reflect the topic of the current work (see analysis_of_references.pdf).

My suggestion for authors is to describe the way they searched and selected literature for the review.

According to the reviewer's suggestion, the way we searched and selected the literature has been included in page 3 (lines 98-108). Thanks.

Reviewer 2 Report

The authors have done an excellent job in characterizing DDR in OC. The review is very comprehensive and timely, and the review is very well written.

Suggestions:

1.       In order to make the reader more intuitive to obtain information, the author should better summary a table to show the molecules (including proteins and miRNA..) involved in DDR.

2.       Detailed mechanism involved in DDR should be summarized/discussed (e.g. the ubiquitination pathways…).

3.       The author should better include a ‘Future Expectations/Perspectives’ part to present their own opinions.

Author Response

The authors have done an excellent job in characterizing DDR in OC. The review is very comprehensive and timely, and the review is very well written.

We thank the reviewer for the positive comments

Suggestions:

  1. In order to make the reader more intuitive to obtain information, the author should better summary a table to show the molecules (including proteins and miRNA..) involved in DDR.

A new table (new Table 2) including the main proteins involved in each DDR pathway has been added to the manuscript. In addition, we have classified the proteins according to their role during the DDR. The expression and function of miRNAs in the DDR are also interesting; however, we consider that this topic would be too long to be included in the present review.

  1. Detailed mechanism involved in DDR should be summarized/discussed (e.g. the ubiquitination pathways…).

The mechanisms involved in the DDR have been summarized at the end of the review, before the conclusions section. In addition, a graphical abstract has been included to visually summarize all the information. New data about the role of ubiquitination in the DDR has also been added to the review (page 11, lines 482-485).

  1. The author should better include a ‘Future Expectations/Perspectives’ part to present their own opinions.

According to the author's suggestion, we have expressed our own opinions in a new "conclusions and future perspectives" section.